

# A Large Contribution of Anthropogenic Organo-Nitrates to Secondary Organic Aerosol in the Alberta Oil Sands

Alex K. Y. Lee[1,2], Max G. Adam[2], John Liggio[3], Shao-Meng Li[3], Kun Li[3], Megan D. Willis[4 #], Jonathan P. D. Abbatt[4], Travis W. Tokarek[5], Charles. A. Odame-Ankrah[5], Hans D. Osthoff[5], Kevin Strawbridge[3],

5    Jeffery R. Brook[3]

[1] Department of Civil and Environmental Engineering, National University of Singapore, Singapore

[2] NUS Environmental Research Institute, National University of Singapore, Singapore

[3] Air Quality Process Research Section, Environment and Climate Change Canada, Toronto, ON, Canada

10    [4] Department of Chemistry, University of Toronto, Toronto, ON, Canada

[5] Department of Department of Chemistry, University of Calgary, Calgary, AB, Canada

[#] Now at Lawrence Berkeley National Lab, Chemical Sciences Division, Berkeley, CA, USA

Correspondence to: Alex K. Y. Lee (ceelkya@nus.edu.sg)



## Abstract

The oil sands industry in Alberta, Canada represents a large anthropogenic source of secondary organic aerosol (SOA). Atmospheric emissions from oil sands operations are a complex mixture of gaseous and particulate pollutants. Their interaction can affect the formation and characteristics of SOA during plume dispersion, but their chemical evolution remains poorly understood. Oxidative processing of organic vapours in the presence of $NO_x$ can lead to particulate organo-nitrate (pON) formation, with important impacts for the SOA budgets, the nitrogen cycle and human health. We provide the first direct field evidence, from ground and aircraft-based real-time aerosol mass spectrometry, that anthropogenic pON contributed up to half of SOA mass that was freshly produced within the emission plumes of oil sands facilities. Using a top-down emission rate retrieval algorithm constrained by aircraft measurements, we estimate the production rate of pON in the oil sands region to be ~15.5 tonnes/day. We demonstrate that pON formation occurs via photooxidation of intermediate-volatility organic compounds (IVOCs) in high $NO_x$ environments, providing observational constraints to improve current SOA modelling frameworks. Our ambient observations are supported by laboratory photooxidation experiments of IVOCs from bitumen vapours under high $NO_x$ conditions, which demonstrate that pON can account for 30-55% of the observed SOA mass depending on the degree of photochemical aging. The large contribution of pON to freshly formed anthropogenic SOA illustrates the central role of pON in SOA production from the oil and gas industry, with relevance for other urban and industrial regions with significant IVOC and $NO_x$ emissions.



## 1. Introduction:

Organo-nitrates (ON), a class of compounds containing the $RONO_2$ and $RO_2NO_2$ moieties, comprise 5–77% (by mass) of ambient organic aerosol in North America and Europe (Kiendler-Scharr et al., 2016;Ng et al., 2017). Particle-phase ON (pON) are formed through chemical reactions between volatile organic compounds (VOCs), $NO_x$ and atmospheric oxidants, with a strong influence on regional $NO_x$ budgets, tropospheric $O_3$ production and atmospheric oxidising capacity (Lelieveld et al., 2016;Liang et al., 1998;Perring et al., 2013). pON can be transported over long distances and act as a source of $NO_x$ in remote locations through gas-particle repartitioning and heterogeneous chemistry (Fry et al., 2013;Liu et al., 2012). Deposition of particulate reactive nitrogen can lead to adverse ecological consequences in nitrogen-limited ecosystems (Matson et al., 2002). Although little is known about the health impacts of pON, inhalation of nitrating reagents in aerosol is thought to be associated with various health risks such as triggering of immune responses and promoting the genesis of allergies (Poschl, 2005). Furthermore, pON can be highly functionalized, which can enhance new particle formation and secondary organic aerosol (SOA) growth (Ehn et al., 2014;Lee et al., 2016), with strong impacts on air quality and climate (Hallquist et al., 2009;Kanakidou et al., 2005).

Many laboratory studies have shown that pON can be produced by photochemical (OH radical initiated) and nocturnal ($NO_3$ radical initiated) oxidation of biogenic and anthropogenic SOA precursors (Lim and Ziemann, 2005;Ng et al., 2017). Extensive field investigations demonstrate the significant formation of pON via nocturnal $NO_3$ radical chemistry of biogenic VOCs on a global scale (Kiendler-Scharr et al., 2016;Ng et al., 2017). Daytime production of pON has also been observed in urban and forested regions (Farmer et al., 2010;Kiendler-Scharr et al., 2016;Lee et al., 2016;Lee et al., 2015b) but the potential role of anthropogenic VOCs in pON formation remains largely unexplored. Until recently, a field study





demonstrated that oil and natural gas drilling operations were associated with alkane-derived pON production (Lee et al., 2015b). Heavier saturated alkanes were more important contributors to pON formation from OH radical oxidation during the daytime, whereas pON from lighter biogenic VOCs dominated nighttime production due to $NO_3$ and $N_2O_5$ chemistry (Lee et al., 2015b). Despite advances in

our understanding of pON, there remains limited direct field observations that (1) evaluate the significance of daytime formation of anthropogenic pON, (2) constrain the contribution of pON to anthropogenic SOA in chemical transport models, and (3) identify the major anthropogenic pON precursors in urban environments and regions that are heavily influenced by large-scale industrial and urban emissions.

Unconventional forms of oil production have become an increasingly important source of oil over the past several decades, with the largest oil sands reserve being found in Alberta, Canada (Alberta Energy Regulator, 2014). The Alberta oil sands region has been recognized as a large source of SOA, $NO_x$, and gas-phase hydrocarbons with a wide range of volatilities (Li et al., 2017;Liggio et al., 2016;Simpson et al., 2010), and hence the potential exists for oil sands operations to be a significant regional source of

pON. Here we present the first direct observational evidence from ground and aircraft measurements that precursor emissions from the Alberta oil sands operations result in the formation of substantial amounts of pON, contributing a significant fraction of freshly formed SOA due to the photooxidation of Intermediate-volatility organic compounds (IVOCs, saturation concentration ($C^*$) = $10^3-10^6$ µg m$^{-3}$) under high $NO_x$ conditions. Both ambient and laboratory measurements illustrate that the observed pON

production and the relative importance of pON to the freshly formed SOA depends upon the degree of photochemical aging in the polluted atmosphere. Our field observations are consistent with our photo-oxidation flow tube experiments exploring pON formation from IVOCs released from bitumen vapours as SOA precursors. Recent modelling work has shown that IVOCs have large impacts on anthropogenic



SOA production (Eluri et al., 2017) and global SOA budgets (Hodzic et al., 2016). Our findings highlight the important role of daytime pON formation for SOA production in urban and industrial regions with strong emissions of anthropogenic IVOCs and $NO_x$ (e.g. fossil fuel combustions), and not only limited to those associated with oil and gas production.

## 2. Experimental Method

### 2.1 Ground-based measurements

An Aerodyne soot-particle aerosol mass spectrometer (SP-AMS) was deployed between August 11 and September 10, 2013 at the Air Monitoring Station 13 (AMS13) ground site managed by the Wood Buffalo

Environmental Association near to Fort MacKay (57.1492° N, 111.6422° W, ~ 270 asl) which is located in the scarcely populated Athabasca oil sands region. The laser-off and -on modes of SP-AMS measurements were used for quantifying the non-refractory particulate matter (NR-PM, including $SO_4^{2-}$, $NO_3^-$, $NH_4^+$ and organics) and refractory black carbon (rBC), respectively (Lee et al., 2015a;Onasch et al., 2012) and for performing source apportionment analysis of ambient organic aerosol via positive matrix

factorization (PMF) (Ulbrich et al., 2009;Zhang et al., 2011). Single-particle characterization was achieved from Aug 22 to 29, 2013 by deploying another co-located SP-AMS equipped with a light-scattering module (Lee et al., 2015a) in order to assess the mixing state of the NR-PM during the ground-based campaign. The SP-AMS was calibrated by size selected ammonium nitrate and Regal Black particles. Further details regarding the calibration and operation of SP-AMS, the quantification of NR-PM

and rBC and the PMF analysis of organic fragments are provided in the supplementary information.

The SP-AMS can detect total nitrate which is the sum of inorganic nitrate and organo-nitrate (ON). Electron impact ionization of nitrate functional groups (-$ONO_2$ and -$O_2NO_2$) in ON generates $NO^+$ and

$NO_2^+$ fragments. A significantly higher $NO^+/NO_2^+$ ratio in the ambient measurements compared to that obtained from pure ammonium nitrate (i.e., a calibration standard) can be used an indicator as substantial contribution of ON in the observed organic aerosol. The mass concentrations of pON and inorganic nitrate ($NO_3^-$) can be estimated based on the observed $NO^+/NO_2^+$ ratio, assuming that the molecular weight and

$NO^+/NO_2^+$ ratio of ON are 200–300 g/mol and 5–10, respectively, based on the method previously described in Farmer et al. (2010) and Xu et al. (2015). The details (i.e. equations and assumptions) of such calculations are provided in the supplementary information.

The AMS13 site provides a comprehensive suite of meteorological and gas-phase measurements of which

wind direction, wind speed, temperature, ozone ($O_3$), sulfur dioxide ($SO_2$), and nitrogen dioxides ($NO_x$) were used in the data analysis. Mixing ratios of $NO_x$ were also quantified by cavity ring down spectroscopy (CRDS) and total odd nitrogen ($NO_y$) was measured using a Thermo Scientific 42i gas analyser. The data was used to determine the -log($NO_x/NO_y$) ratio for estimating the relative age of air masses. The mixing height data was obtained from a dual wavelength backscatter lidar which is in continuous and autonomous

operation except during periods of precipitation. For details on the data being used in this study, please refer to the supplementary information and Tokarek et al. (2018). The bivariate polar plots to illustrate the variation of the concentration of species with wind speed and wind direction were generated using the openair R package and are described in more detail in the supplementary information.

**2.2 Aircraft measurements**

Flights investigating the transformation of oil sands pollutants downwind of facilities were conducted during the same period as the ground-based measurements. The nature of the flights, and the use of TERRA to derive emissions for various pollutants has been described in great detail in numerous



publications (Gordon et al., 2015;Li et al., 2017;Liggio et al., 2016). Briefly, transformation flights were designed as Lagrangian experiments and flown as virtual screens, such that plumes were repeatedly sampled at different times downwind with no industrial emissions between the screens. In the current work, a high-resolution time-of-flight aerosol mass spectrometer (HR-ToF-AMS) was used to measure

5    the chemical composition of ambient aerosol particles and the formation of SOA and pON over time is investigated using primarily Flight 19 (F19) as it was the most successful Lagrangian experiment, having the best agreement between air parcel transport times and aircraft flight times. pON mass was determined in the same manner as for the ground site measurements, using a $NO^+/NO_2^+$ ratio of 3.5±1.5 as determined during HR-ToF-AMS calibrations.

The TERRA algorithm determines the transfer rate of pollutants through the walls of a virtual screen or box. The transfer rate is derived using the Divergence Theorem (Gordon et al., 2015). After accounting for horizontal and vertical advection and turbulence, and air density changes, the transfer rate is equivalent to the emission rate of the pollutant. In the current case, the SOA formation rate is the difference in organic

15   aerosol transfer rate between any two screens, assuming there are no other primary emissions between screens and ignoring dry deposition. The total pON (or SOA) production rate is taken to be the transfer rate through the final screen and since dry deposition is not accounted for is considered to be a lower limit to the actual ON formation rate. The spatial extent of the oil sands plume (including the pON and SOA formation within such plumes) across any given screen is defined using the spatial extent of BC as a

20   surrogate, as it is known to originate from mining activities in the oil sands, as described in Liggio et al. (2016). Liggio et al. (2016). The total pON production rate over the course of the flight (4 hours) is extrapolated to a photochemical day (i.e., tonnes/day) via scaling to the OH radical, as described previously (Liggio et al., 2016).



### 2.3 Laboratory flow tube experiments

A laboratory study of SOA formation from bitumen vapour was conducted using a 1L flow tube reactor, which has been described in detail elsewhere (Liu et al., 2014). Briefly, precursors derived from bitumen

ore were oxidized by OH radical in the presence of $NO_x$, simulating up to several hours of oxidation. The OH radicals inside the reactor were generated by photolysis of 6 ppm $O_3$ at 254 nm in the presence of water vapour (~35% RH). The total flow rate was 0.8 L min$^{-1}$ resulting in a reaction time inside the reactor of 75 s. A bitumen ore sample obtained directly from an active mine was placed in a glass U-tube to which a small flow of zero air was added and introduced into the flow tube. The total gas-phase organic carbon

(TOC) concentration was estimated by converting all carbon to $CO_2$ using a Pd catalyst and measuring the resultant $CO_2$ concentration (Veres et al., 2010). The TOC in the flow tube was $375 \pm 30$ ppbC, corresponding to $25 \pm 2$ ppb of organic species when assuming an average carbon number of 15 (Liggio et al., 2016). The OH concentration was adjusted by changing the intensity of the 254 nm UV lamp. The OH exposure ranged from $1.11 \times 10^{11}$ to $3.06 \times 10^{11}$ molecule cm$^{-3}$ s, which corresponds to 4.3-12.2 hours

when assuming a daytime OH concentration of $7 \times 10^6$ molecule cm$^{-3}$ as estimated for polluted areas (Hofzumahaus et al., 2009;Stone et al., 2012) and consistent with estimations for oil sands plumes (Liggio et al., 2016).

As NO was consumed very quickly under ppm level $O_3$ within the flow tube, traditional methods of NO

addition to the system are not ideal for high-$NO_x$ experiments. Hence, percent level $N_2O$ (2%) was used to generate NO. $N_2O$ reacts with $O(^1D)$ (which is formed by photolysis of $O_3$) to generate NO: $N_2O + O(^1D) \rightarrow 2NO$. This approach can provide a relatively high NO concentration uniformly throughout the entire reactor, and has proven to be a reliable approach for performing high-$NO_x$ experiments using flow



tube reactors (Lambe et al., 2017). Using this method, we can achieve a NO concentration of 2-12 ppb and a $NO_2$ concentration of 20-120 ppb. The aerosol composition in the flow tube was measured by a Long-TOF-AMS (L-TOF-AMS, Aerodyne Research Inc.). In the high-$NO_x$ experiments, inorganic nitrate and ON were formed, with both of them generating $NO^+$ and $NO_2^+$ fragments. Hence the inorganic and

organic nitrate must be distinguished from each other, and the total N signal was determined by following the approach proposed by Farmer et al. (2010) (see supplementary information).

## 3. Results and Discussion:

### 3.1 Significant contribution of pON to SOA in oil sands plumes

The chemical compositions of NR-PM and rBC was measured using a SP-AMS at a ground site (Figure S1). Three types of organic aerosol (OA), referred to as hydrocarbon-like OA (HOA), less-oxidized oxygenated OA (LO-OOA) and more-oxidized OOA (MO-OOA) were identified based on positive matrix factorization (PMF) of organic fragments (Figures 1, S2 and S3). In this work, we focus on the origins and chemical characteristics of LO-OOA, which represents fresh SOA that was largely associated with

the observed pON concentrations at the ground site as discussed below. Note that MO-OOA represents SOA formed via oxidation of VOCs that were relatively well mixed in the atmosphere rather than those freshly emitted from the oil sands facilities (See supplementary information for the description of HOA and MO-OOA factors).

Strong plumes of gaseous $SO_2$, particulate $SO_4^{2-}$ and LO-OOA were occasionally observed at the ground site, which were primarily transported from the east and southeast directions (Figures 1 and S11), indicating the presence of large anthropogenic sources in the oil sands region. The largest possible source of $SO_2$ and $SO_4^{2-}$ are upgrading facilities in which sulfur and nitrogen constituents are removed from the



bitumen. LO-OOA was moderately correlated with $SO_4^{2-}$ ($r^2 = 0.46$, Table S1), suggesting that this SOA component (or its precursors) also originated from the oil sands facilities and was subsequently mixed with $SO_4^{2-}$ plumes during dispersion. Furthermore, a fraction of LO-OOA was internally mixed with $SO_4^{2-}$ in sulfate-rich plumes, which is supported by single particle measurements and PMF analysis during a

short period (22-29 August 2013) with relatively large influences from oil sands emission plumes (Figures S2-S7, and S10-S12, see supplementary information). These ground-based observations are in good agreement with the aircraft-based high-resolution aerosol mass spectrometry measurements over the same oil sands region. Two major types of plumes were frequently observed, dominated by either $SO_4^{2-}$ or fresh SOA (i.e., > 90% of total NR-PM mass) (Liggio et al., 2016). The mass spectrum of fresh SOA-rich

plumes observed on the aircraft (Liggio et al., 2016) was similar to that of LO-OOA observed at the ground site (Figure S3), confirming the anthropogenic origin of LO-OOA.

The LO-OOA component observed in oil sands plumes was strongly correlated with the total signals for nitrate fragments (i.e., $NO^+$ and $NO_2^+$, $r^2 = 0.88$, Table S1). AMS-measured $NO^+$ and $NO_2^+$ signals can

originate from both pON and inorganic nitrate. A significantly higher $NO^+/NO_2^+$ ratio observed at the ground site compared to that of pure ammonium nitrate (Figure S8) indicates that pON contributed substantially to the observed $NO^+$ and $NO_2^+$ signals. The mass concentrations of pON and inorganic nitrate ($NO_3^-$) were estimated based on the observed $NO^+/NO_2^+$ ratio at the ground site, assuming that the molecular weight and $NO^+/NO_2^+$ ratio of ON are 200–300 g/mol and 5–10, respectively (Farmer et al.,

2010;Xu et al., 2015). A strong correlation between the mass loadings of pON and LO-OOA ($r^2 = 0.77$) suggests that ON was a component of LO-OOA, contributing approximately 24-53% of LO-OOA mass on average (Figure S9 and Table S3). Mass loadings of pON and $NO_3^-$ calculated from this approach are shown in Figure 1b.

Aircraft measurements were designed to trace the chemical evolution of SOA within plumes downwind of oil sands facilities (Liggio et al., 2016). A higher average value of $NO^+/NO_2^+$ ratio of SOA-rich plumes compared to that of pure ammonium nitrate was also observed in our aircraft measurements, further

demonstrating a large contribution of pON to SOA. The pON mass derived from these aircraft measurements correlated well with anthropogenic SOA that was freshly formed during plume dispersion ($r^2 = 0.71$, Figure 2a and 2b), and accounted for up to ~34 % (±18%) of such SOA by mass (Figure S13). Utilizing the top-down emission rate retrieval algorithm (TERRA) (Gordon et al., 2015) and the aircraft-based aerosol mass spectrometry measurements, we derived a significant pON production rate of ~15.5

tonnes/day (or 1.2 tonnes/h) (Figure 2c). Together, our ground and aircraft-based measurements provide direct field evidence for a strong association of pON with SOA formation chemistry in the emission plumes of oil sands facilities.

**3.2 Daytime pON formation and IVOCs as potential precursors**

The diurnal patterns of LO-OOA and pON suggest their production was driven by daytime photochemistry in the presence of $NO_x$. LO-OOA and pON peaked between 10:00–13:00 local time, in conjunction with the higher levels of $NO_x$ (8–13 ppb on average), and had relatively low concentrations during the nighttime (nighttime $[pON]_{avg}$ = 0.4–0.6 µg/m$^3$ vs. daytime peak $[pON]_{avg}$ = 0.9–1 µg/m$^3$, Figure 1j and 1k). Mass concentrations of pON in some of the major emission plumes ranged from 2.2 to 5.3 µg/m$^3$ (Figure 1b).

Tokarek et al. (2018) performed principle component analysis of 28 co-located measurements at the ground site and reported that LO-OOA was strongly associated with unresolved IVOCs measured by gas chromatograph-ion trap mass spectrometry (GC-ITMS), while biogenic VOCs including α-pinene, β-pinene and limonene were not associated with LO-OOA. These results suggest that tailings ponds, mine

fleet and vehicle emissions, and mining and processing of raw bitumen are the major sources of IVOCs in the oil sands region. Furthermore, Liggio et al. (2016) illustrated that substantial amounts of fresh SOA could be generated from SVOCs and IVOCs emitted from oil sands operations (i.e., 86% of the SOA observed at Screen A of Figure 2c), dominating over SOA from total traditional biogenic and

anthropogenic VOC precursors ($C^* > 10^6$ μg/m$^3$). These observations suggest that IVOCs were the major precursors of pON observed in the oil sands region.

Laboratory flow tube experiments were conducted to investigate pON formation through photooxidation of bitumen hydrocarbon vapours under high NO$_x$ conditions. The volatility profile of organic vapours

evaporated from bitumen ore has been previously reported (Liggio et al., 2016). Bitumen vapours are dominated by species with intermediate volatility (IVOCs, i.e., range from C$_{11}$–C$_{17}$ of n-alkane and peak at C$_{13}$–C$_{14}$) and are consistent with the IVOC volatility profiles observed at the ground site during polluted periods (Liggio et al., 2016;Tokarek et al., 2018). Based on the results from aerosol mass spectrometry measurements of the flow tube experiments, pON contributed approximately 30–55% of the total SOA

mass over an estimated photochemical age of 4–12 h, assuming a daytime OH concentration of $7 \times 10^6$ molecule cm$^{-3}$ typical for polluted areas (Hofzumahaus et al., 2009;Stone et al., 2012) and consistent with estimates for oil sands pollution plumes (Liggio et al., 2016). The fractional contribution of pON to SOA determined from the flow tube experiments were in agreement with our ambient field and aircraft observations (Figure 3c).

Hydrocarbons, including alkanes, alkenes, and aromatics, accounted for > 90% of SOA precursors emitted from oil sands facilities (Li et al., 2017). Previous laboratory studies and model simulations demonstrate that photooxidation of linear, branched and cyclic hydrocarbons in the presence of NO$_x$ can produce pON,



such as alkylnitrates and hydroxynitrates (Jordan et al., 2008;Lim and Ziemann, 2009;Matsunaga and Ziemann, 2010). The SOA formation mechanisms and product distributions from such reactions may differ due to molecular structure of precursors, which in turn impacts pON formation potential. For example, Matsunaga and Ziemann (2010) demonstrated that yields of pON due to OH radical oxidation in the presence of $NO_x$ increased with the carbon number of 2-methyl-1-alkenes (from $C_9$ to $C_{15}$), primarily due to enhanced gas-to-particle partitioning, and reached a plateau for $C_{14}$-$C_{15}$ precursors, which fall into the IVOCs range of hydrocarbons observed in the bitumen vapour as discussed above.

### 3.3 Decreasing contribution of pON to SOA with photochemical aging

Using $-\log(NO_x/NO_y)$ as a proxy of the average photochemical age (PCA) of air masses and the organic-to-rBC ratio as an indication of SOA formation, observations at the ground site indicate that SOA concentrations increased continuously and the overall OA became more oxygenated with increasing PCA up to approximately 4 h (Figure 3a). Of particular interest is that the rBC-normalized mass concentrations of pON (and LO-OOA) increased with PCA only when PCA was ≤ 1 h, becoming roughly constant thereafter (Figure 3b). This observation provides evidence for freshly formed SOA containing pON. The average pON-to-LO-OOA ratios in fresh SOA plumes at the ground site were up to ~ 0.6 (red circles in Figure 3c). Relatively low $NO_x$ levels might limit pON production when the estimated PCA > 1 h (Figure 3d, i.e., at times without strong influence from $NO_x$ emissions). Although the lifetime of anthropogenic pON remains poorly understood, recent laboratory and modelling studies have shown that a short atmospheric lifetime of biogenic VOC-derived pON, on the order of hours, could be due to different chemical loss mechanisms (Boyd et al., 2015;Pye et al., 2015;Zare et al., 2018). Substantial evaporative loss (i.e., LO-OOA generally represents a more volatile fraction of OOA), hydrolysis and photo-



degradation of pON when both temperature and photochemical activity were high could limit pON production at longer PCA (Figure 1k).

Our aircraft-based measurements show that both $\Delta SOA/\Delta rBC$ and $\Delta pON/\Delta rBC$ ($\Delta$ represents the change
in the plume observations after background subtraction) increased continuously as a function of PCA within emission plumes (from flight screen A to D, Figure 3a and b). This observation confirms that both SOA and pON were freshly formed during their transport up to a PCA of 5-6 h (i.e., extending our observations from the ground site to larger PCA). One of the possible reasons for more sustained pON production in the aircraft observations is that the $NO_x$ mixing ratios, and potentially IVOCs concentrations, remain relatively high within the plumes compared to at the ground site (Figure 3d), which was not continuously impacted by plumes. Nevertheless, the contribution of pON to fresh SOA mass decreased with PCA from 0.49 to 0.31 (i.e., further downwind of the oil sands facilities, Figure 3c). Such decreasing trend could be due to decreasing mixing ratios of $NO_x$ (Figure 3d) in the plume caused by dilution downwind of the emission source, which may make the formation of non-pON fresh SOA and multi-generation products becoming more important in the later stage of oxidative process, and larger degree of degradation and evaporative loss of pON as a function of PCA compared to other types of SOA products.

A decreasing trend in pON-to-SOA ratio (from 0.55 to 0.30) was also observed with PCA in the flow tube experiments (Figure 3c). However, $NO_x$ was not the limiting factor for pON production. The $NO_x$ mixing ratios in the flow tube were much higher than those observed from the aircraft, and led to higher pON-to-SOA ratios for the flow tube SOA within the similar range of PCA (Figure 3c and 3d). The flow tube experiments were conducted at a relatively constant RH (~35%) with a fixed residence time of 75 s, so that particle-phase hydrolysis is unlikely the governing factor of the decreasing trend. Overall, in addition



to the effects of NO$_x$ mixing ratio and hydrolysis of pON, both aircraft observations and laboratory flow tube experiments highlight the importance of investigating relative contributions of anthropogenic pON and other SOA products as a function of PCA, especially under atmospherically relevant conditions that allow multi-step oxidation of precursors and intermediate products.

## 4. Conclusions and Atmospheric Implications

Ambient observations involving a comprehensive suite of ground and aircraft-based measurements in the Alberta oil sands region, combined with laboratory flow tube experiments, have shown that daytime production of anthropogenic pON can contribute up to ~50% of fresh SOA. Utilizing aircraft data and

10    TERRA algorithm, we estimate a pON production rate of 15 tonnes/day from the Alberta oil sands region. Given the recent observation that oil sands operations in Alberta can be one of the largest anthropogenic sources of SOA in North America (45–84 tonnes/day) by comparing to the estimated SOA production rates in different cities (Liggio et al., 2016), the large contribution of pON to SOA in the pollutant plumes highlight the potential of oil sands operations as a significant source of anthropogenic pON.

Despite the Alberta oil sands region being largely vegetated, both ground- and aircraft-based observations suggests that nocturnal chemistry of NO$_3$ radicals and biogenic VOCs was not the key formation mechanism of pON in the Alberta oil sands during our field campaign. Rather, this work provides direct evidence for a significant contribution of pON to fresh anthropogenic SOA, driven by photochemistry and

20    emission of IVOCs and NO$_x$ from large-scale industrial facilities. Anthropogenic IVOCs can form SOA with much higher yields (5 times) compared to other single-ring aromatic VOCs (Zhao et al., 2014). There is increasing evidence that vehicular emissions can be a significant source of IVOCs in urban environments, leading to substantial anthropogenic SOA production (Liu et al., 2017;Zhao et al.,



2014;Zhao et al., 2015). Including IVOCs in SOA prediction models can have great impact on estimating the production of anthropogenic SOA in urban environments and global SOA budgets (Eluri et al., 2017;Hodzic et al., 2016). However, anthropogenic pON formation chemistry has not been fully integrated into current SOA prediction models. Our findings highlight the significance of investigating the role of

pON formation in SOA production in other urban and industrial regions with strong emissions of anthropogenic IVOCs and $NO_x$.

Both aircraft observations and flow tube experiments demonstrate that the mass fraction of pON in fresh SOA decreases as a function of PCA of the air masses. However, detailed anthropogenic pON formation

and degradation mechanisms, and the lifetime, remain poorly understood. More research is required to improve our understanding on the sources and sinks of anthropogenic pON for constraining chemical transport models as well as their subsequent environmental implications. pON accounted for up to 21% of total OA mass in the Alberta oil sands region, which is comparable to other locations worldwide (Kiendler-Scharr et al., 2016;Ng et al., 2017). In contrast, the average mass fraction of organic $-ONO2$

functionality to total particulate nitrate (i.e., organic $-ONO_2 + NO_3$) was 0.83, which is much higher than the average (0.33) reported in North America and Europe (Kiendler-Scharr et al., 2016;Ng et al., 2017). Given that pON can be converted back to NOx during atmospheric transport, the large contribution of organic $-ONO_2$ to total particulate nitrate may have significant implications for the predictions of nitrogen deposition (including nitrogen from $NO_2$ and total particulate nitrate) from the industrial center to the

surrounding boreal forest ecosystem and regions (Fenn et al., 2015;Hsu et al., 2016).

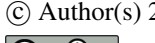



**Author contribution**

AKYL prepared the manuscript with contributions from all co-authors. AKYL, MDW, TWT, CAO-A, and HDO operated the instruments in the field and analyzed resulting data. MGA analyzed the field data. AKYL, MDW, JPDA and JRB designed the field experiment. JL and S-ML designed, conducted and

5    analyzed the aircraft measurements. JL and KL designed and conducted the laboratory experiments. KS analyzed mixing height data.

**Acknowledgement**

Funding for this work was provided by the Natural Sciences and Engineering Research Council of Canada

10    (NSERC), Environment and Climate Change Canada, and the Oil Sands Monitoring Program (JOSM).



**Figures and Captions**

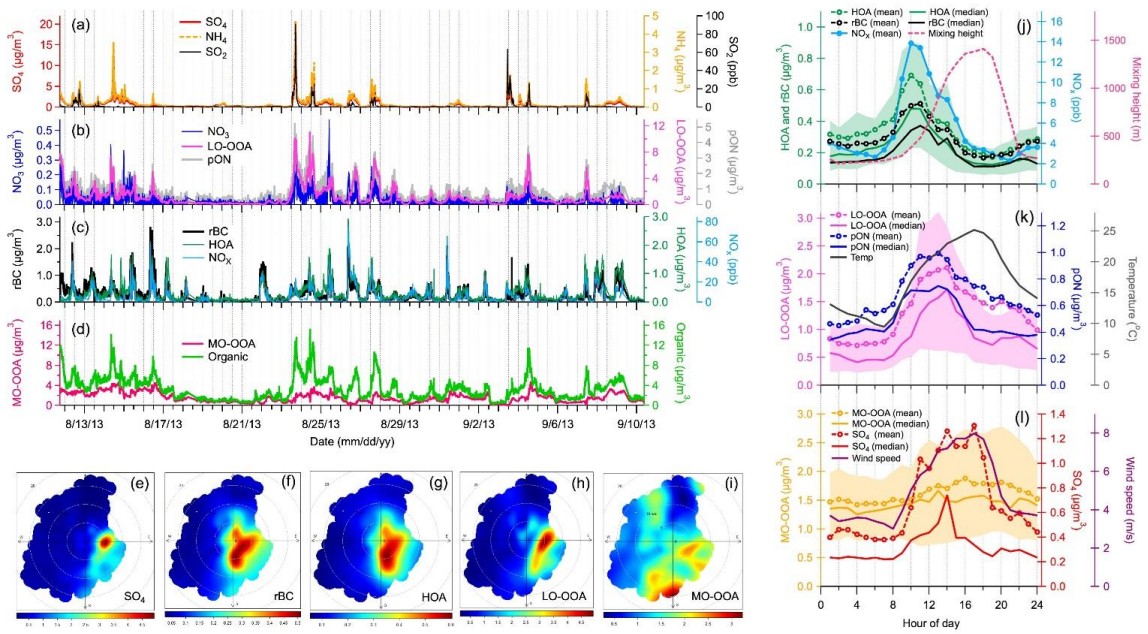

5     **Figure 1.** Ground-based observations: Time series (a-d) and wind rose plots (e-i) of gas-phase species ($SO_2$ and $NO_x$), particle-phase species (Organics, $SO_4$, $NO_3$, $NH_4$ and rBC) and PMF factors (HOA, LO-OOA and MO-OOA). Correlations ($r^2$ values) between all these measurements are presented in Tables S1 for the entire sampling period. Other wind rose plots are shown in Figure S11. (j-l) Diurnal patterns of SP-AMS measurements (rBC and $SO_4$), PMF factors (HOA, LO-OOA and MO-OOA), organo-nitrate (pON),

10     $NO_x$, ambient temperature, local wind speeds and mixing height. While mean values could reflect the influences of pollutants plumes on mass loadings, median values could better represent the central tendency of measurements that are less affected by individual plumes. Both mean and median values of SP-AMS measurements and PMF factors are presented. The upper and lower values of shaded regions represented 25 and 75 percentiles of diurnal variations of PMF factors, respectively.



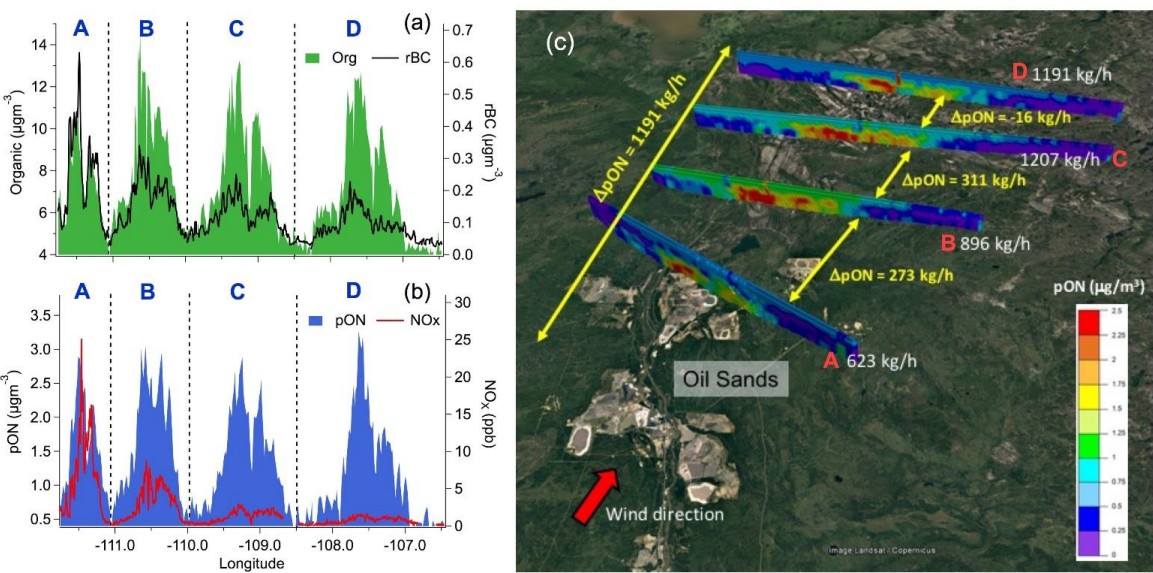

**Figure 2.** Aircraft observations: (a) Organic aerosol and rBC (b) pON and NO$_x$ concentrations measured
at the four screens (A to D) shown in panel (c). Organic aerosol and pON concentrations remained roughly
constant during dispersion, whereas rBC and NO$_x$ decreased continuously. (c) Results from the TERRA
algorithm, which estimate a total pON production of 1.2 tonnes/h (or 15.5 tonnes/day) downstream of oil
sands operations (i.e., the sum of ΔpON between the four screens A to D), constrained by the aircraft
measurements.




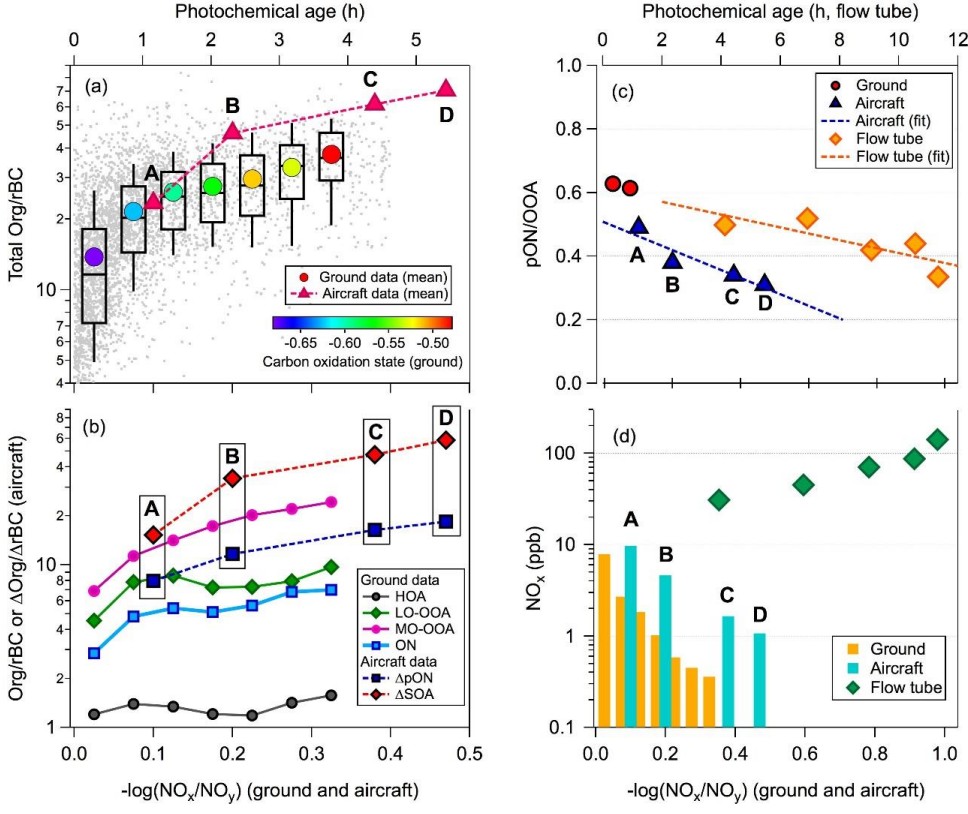

**Figure 3.** Comparison between ground-based, aircraft and flow tube data as a function of $-\log(NO_x/NO_y)$
or photochemical age (PCA): (a) rBC normalized mass loadings of total organic, (b) rBC-normalized mass
loadings (mean) of HOA, LO-OOA, MO-OOA and pON (c) pON/OOA ratios (i.e., OOA = LO-OOA
observed from the ground site, OOA = total SOA in the plume observed from the aircraft measurement,
and OOA = total SOA produced in the flow tube experiments) and (d) $NO_x$ mixing ratio. The pON data
presented here are upper limit based on the assumption of $R_{ON} = 5$ and $MW_{ON} = 300$ g/mol. The colour
scale of solid circles in panel (a) represents the average carbon oxidation state of OA materials Assuming
ambient daytime OH radical concentration was ~7 x $10^6$ molecules/cm$^3$ (Liggio et al., 2016) and the major
$NO_x$ loss product was $HNO_3$. The PCA of air masses (i.e., PCA = - ln([$NO_x$]/[$NO_y$]) / $k_{rxn}$ [OH]) were
estimated using a rate constant between OH radical and $NO_x$ for $HNO_3$ formation ($k_{rxn}$) of 7.9 x $10^{-12}$ cm$^3$
molecules$^{-1}$ s$^{-1}$ (Brown et al., 1999;Cappa et al., 2012). Only the first two points of ground data were
included in panel (c) as a reference for the pON/OOA ratio of the fresh SOA. The symbols of A to D in
all the panels refer to the measurements at the four screens indicated in Figure 2c.



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
