# Peer review of "A Large Contribution of Anthropogenic Organo-Nitrates to Secondary Organic Aerosol in the Alberta Oil Sands"

_Atmospheric Chemistry and Physics, 2018_

## Short Comment (SC1) · 16 Jan 2019

Very nice work! In pondering the apparent decrease in the pON content (and relative fraction of SOA) with PCA that you report, I was reminded of another recent paper that is still in the open discussion (I think). The paper by Schwantes, et al. https://doi.org/10.5194/acp-2018-1358 reports the formation of organic dinitrates (among other products) during the high-NOx photooxidation of isoprene and suggests (section 5.3) a negative measurement bias in the AMS determination of these compounds (and perhaps all pON). It isn't clear to me that the methods used to determine pON are the same in both papers, but it begs the question of whether some sort of in-

terference could be affecting the measurements of the highly oxidized oil sands SOA, perhaps leading to a lower effective CE for the more oxidized dinitrates. Perhaps this could account for the apparent decrease in the absolute and relative contribution of the nitrated products.

---

## Referee Comment (RC1) · Anonymous Referee #1 · 8 Feb 2019

Lee et al. present an analysis of the contribution of organo-nitrates to secondary organic aerosol formation from the Alberta oil sands based on field observations and laboratory experiments. The analysis is generally insightful, and the manuscript represents useful addition to our understanding of the role of IVOCs and NOx in the formation of aerosols from oil and gas producing regions; however, the reasoning behind some steps in the analysis is not clearly explained. I would recommend publication after the following comments are addressed.

General Comments:

1. In several places throughout the manuscript, the authors suggest that the results

from this study provide insight about SOA formation in urban environments. However, neither the field study nor the laboratory experiments appear to specifically probe urban SOA formation, thus the connection to urban SOA appears to be based primarily on the fact that urban areas also have IVOCs and NOx. I feel further analysis of likely IVOC composition in cities and their relation to those produced from oil and gas production regions is necessary to support this argument.

2. The discussion of the laboratory experiments would be helped by some additional details about the reactions likely happening in the flow tube. Specifically:

a) Can any estimates be made about the possible fate of RO2 radicals? Given the high concentration of OH and (relatively) moderate concentration of NO, I worry that RO2-RO2 reactions will be far more prominent than they would be under ambient conditions. If this is the case, the SOA yields from the flow-tube would not be representative of the yields from atmospheric oxidation.

b) Is there any possibility that NO3 reactions are occurring? I think this is unlikely given the presence of NO and of UV light, but with ppm levels of O3, it feels like a possibility.

3. In section 3.3, the loss of pON is suggested several times as a possible or likely mechanism to explain the results of Fig. 3. Based on the combined measurements from the ground, the aircraft, and the laboratory, I am not convinced that there is evidence for pON loss instead of limited pON production at longer timescales. The aircraft and ground data presented in Fig. 3b show an increase in pON concentration with increasing PCA. Furthermore, Lee et al., 2015b found that pON concentrations at another oil and gas producing region could be well described with no additional loss of pON besides turbulent mixing. To me, this suggests that the apparent decrease in pON with time is due entirely to the normalization to total SOA concentration.

More broadly, I would appreciate further discussion about what is gained by framing the results in terms of the fraction of pON or LO-OOA to total SOA, rather than examining the concentration of pON or LO-OOA directly. Put differently, does examining the

decreasing trend in pON-to-SOA tell us anything about the production or loss of pON, or does it tell us something about the production of SOA as a whole?

Specific Comments:

Page 3, Line 16-21: I think the results of Lee et al., 2015b should be given more consideration, as that paper found that daytime oxidation of anthropogenic long-chain hydrocarbons was a major source of pON in an oil and gas producing region.

Page 5, Line 3-4: See general comment 1 above. I don't feel the connection to urban regions has been adequately made in this manuscript to make this claim.

Page 5, Line 11: For those unfamiliar with this region, could you clarify the relation between the Alberta oil sands and the Athabasca oil sands?

Page 9, Line 13-18: The association of different PMF factors with different sources plays a large enough role in this analysis that I think a greater discussion of the different PMF factors and their likely sources would be appropriate to include in the main text.

Page 11, Line 6: The description here describes pON mass being well correlated with freshly formed anthropogenic SOA, while Fig. 2a shows total organic aerosol - is the assumption that all the SOA measured is freshly-formed and anthropogenic, or was only a one SOA factor used in the correlation?

Page 11, Line 23: I parse this sentence as saying that because LO-OOA was strongly associated with unresolved IVOCs, the IVOCs must be produced by oil and gas extraction activities, which does not make sense to me. This reasoning should be clarified or an alternate justification for the likely sources of IVOCs should be provided.

Page 12 Line 21 - Page 13, Line 7: I am having trouble understanding how this paragraph fits into the chain of reasoning in section 3.2 The previous paragraphs form a seem to follow a logical progression, setting up oil-sands IVOCs as the likely precursor for pON and confirming with flow tube experiments. I would suggest clarifying the goal of this paragraph, or moving it to a different section in the text.

Page 13, Line 10-11: How does the analysis of photochemical age for the ground measurements distinguish between aging of an isolated airmass and mixing of airmasses of different ages from different sources. Is this what the normalization to black carbon was used for? If so, this should be stated explicitly in the text.

Figure 3: It might be useful to include in this figure or in the supplement how the absolute concentration of pON changed with photochemical age in the flow tube experiment. These results might be able to help clarify whether loss of pON was being observed.

Page 13 Line 21 - Page 14 Line 2: This sentence appears to equate the evaporation or degradation of pON with changes to pON production. This should be reworded to clarify that the production and loss terms of pON can change independently. I am also confused by the reference to Figure 1k, as that figure does not directly reference photochemical age at all.

Page 15, Line 11-14: Could this sentence be re-worded? I am not sure that it is grammatically correct and it took me several attempts to understand its meaning, that the production of pON from the Alberta oil sands is significant on regional or continental scales.

Technical Corrections:

Page 3, Line 24: The last sentence on this page ('Until recently....') is not grammatical.

Page 7, Line 21: The citation 'Liggio et al. (2016)' appears twice.

Page 14, Line 14-16: This clause ('which may make the formation of non-pON fresh SOA') is missing an object.

---

## Referee Comment (RC2) · Anonymous Referee #2 · 15 Feb 2019

The manuscript presents ground and flight-based observations of particulate organic nitrates and total organic aerosol mass concentrations in the summer-fall season in/near the Athabasca oil sands region. According to the calculations presented, pON contributes as much 55% of OA mass concentration in freshly emitted airmass, but that contribution decreases with photochemical age. This reviewer has serious concerns regarding the assumptions made in (1) quantifying pON, and (2) attributing the field observed enhancement in OA to oxidation of bitumen vapors. Major revisions/clarifications are needed prior to publication.

(1) A clearer accounting of pON mass contribution is needed. As the authors note,

figure 3 shows the upper limit contribution of pON by assuming a constant molecular weight of ON of 300 g/mol. How reprentative is this value to ON derived from bitumen vapor oxidation? The range of typical ON molecular weights reported by Farmer et al. & Xu et al. were focused on different parent VOCs. Moreover, a wide range in R values was observed during the campaign, as seen in figure S8. Doesn't this reflect that numerous parent compounds contributed to pON growth, hence, of varying molecular weight of pON? Is this variability in R accounted for in the quantification of pON? If not, show how big an impact the variability in R could have on pON quantification. Is there a trend in the R value with photochemical age? Could the changing R with PCA be responsible for the apparent decreasing pON/OA ratio? Revise figure 3 and numbers reported in the manuscript to clearly show a reasonable range of pON contribution to OA by acknowledging the assumptions made here.

Also, it is noted on line 7-9 page 7 that R = 3.5+/-1.5 was determined by calibrations, but no details of these calibrations are provided, and an R value of 5 is used in the figures. How many different VOCs (aromatics, alkanes, alkenes, etc.) were characterized? What governs the variability in the R values?

(2) The authors use the fact that similar fractional contributions of pON to SOA - that is, between 30 and 55% - were observed in the ambient fresh plumes as in the flow tube experiments as evidence that bitumen vapors were the source of the pON (and SOA). Using such a metric - particularly one with a sizeable range, in a flow tube with unrealistic chemical conditions, where pON contribution to OA appears strongly dependent on photochemical lifetime - as an identifying marker seems highly questionable as it most certainly will not be specific to bitumen vapor oxidation. Have the authors attempted any other VOCs - say isoprene, monoterepenes, or any of the possible emission sources listed at the end of page 11 and start of page 12 - in the flow tube experiments to rule out other VOC sources?

There is an odd sentence in the conclusion, that "pON accounted for 21% of total OA mass, which is comparable to other locations," studies by Kiendler-Scharr et al. and Ng

et al. Where did this 21% come from? Also, both of those studies focued on residential and urban areas. But if pON/OA is comparable regardless of region of study, why should we care about pON from oil sands? Isn't it possible the plumes intercepted by the aircraft had elevated HOx (due to elevated NOx) that rapidly oxidized biogenic VOC entrained into the plume?

The authors conclude the pON is formed largely by daytime chemistry. I would like to see included in figure 1 a diel plot of the fraction of pON to OA.

minor page 10 line 20-22. the 24-53% range, is that accounting for variabililty through campaign assuming a constant molecular weight, or range due to assuming 200-300 g/mol molecular weight?

page 11 line 5-7; this sentence is not supported by the preceding sentence.

are you including the mass of the nitrate functional group when reporting mass of pON or just the organic portion?

page 3 lines 12-14, is that true, that the composition of pON can affect SOA growth and npf? in any case, these are probably not the correct citation.

page 4 lines 19 - 21, need citation

---

## Author Comment (AC1) · 19 Jul 2019

**Point-by-point response**

We would like to thank for the constructive comments from the two reviewers.

**Reviewer 1:**

Lee et al. present an analysis of the contribution of organo-nitrates to secondary organic aerosol formation from the Alberta oil sands based on field observations and laboratory experiments. The analysis is generally insightful, and the manuscript represents useful addition to our understanding of the role of IVOCs and NOx in the formation of aerosols from oil and gas producing regions; however, the reasoning behind some steps in the analysis is not clearly explained. I would recommend publication after the following comments are addressed.

General Comments:

1. In several places throughout the manuscript, the authors suggest that the results from this study provide insight about SOA formation in urban environments. However, neither the field study nor the laboratory experiments appear to specifically probe urban SOA formation, thus the connection to urban SOA appears to be based primarily on the fact that urban areas also have IVOCs and NOx. I feel further analysis of likely IVOC composition in cities and their relation to those produced from oil and gas production regions is necessary to support this argument.

Response: Hydrocarbons are the major gas-phase precursors in the oil sands pollutant plumes as reported in our previous aircraft measurement (Li et al., 2017). The recent literature has shown that hydrocarbons in the range of IVOCs can be largely emitted from vehicular emissions and other petroleum-based sources in urban (Zhao et al., 2014;2015). This provides important connections between the finding of this work and pON formation in urban SOA. The similar argument has been used in Lee et al. (2015b) to highlight the relevance of pON formation observed at another oil and gas production region to urban pON formation. Although we agree that further analyses of IVOC composition between urban and oil and gas production regions are required in the future as IVOC chemical compositions remain largely unspeciated, our results can highlight the importance of investigating pON formation due to IVOCs in other urban settings. The paragraph in the section of "Conclusion and atmospheric implication" has been revised as following.

"Our aircraft measurements have shown that hydrocarbons including alkanes and aromatics are dominant gas-phase precursors within the plumes (Li et al., 2017). There is increasing evidence that IVOCs emitted from vehicle and other petroleum-based sources can lead to substantial anthropogenic SOA production in urban environments (Liu et al., 2017;Zhao et al., 2014;2015). Such urban IVOCs comprise different molecular structures of alkanes and aromatics, which can form SOA with much higher yields (5 times) compared to single-ring aromatic VOCs (Zhao et al., 2014). Including IVOCs in SOA prediction models can have great impact on estimating the production of anthropogenic SOA in urban environments and global SOA budgets (Eluri et al., 2017;Hodzic et al., 2016). However, anthropogenic pON formation chemistry has not been fully integrated into current SOA prediction models. Although IVOCs chemical compositions remain largely unspeciated (~80-90% of total IVOCs) in urban air and vehicle emissions (Zhao et al., 2014;2015), our findings highlight the significance of investigating the role of pON formation in

SOA production in other urban and industrial regions with strong emissions of hydrocarbon in the range of IVOCs"

2. The discussion of the laboratory experiments would be helped by some additional details about the reactions likely happening in the flow tube. Specifically:

a) Can any estimates be made about the possible fate of RO2 radicals? Given the high concentration of OH and (relatively) moderate concentration of NO, I worry that RO2-RO2 reactions will be far more prominent than they would be under ambient conditions. If this is the case, the SOA yields from the flow-tube would not be representative of the yields from atmospheric oxidation. b) Is there any possibility that NO3 reactions are occurring? I think this is unlikely given the presence of NO and of UV light, but with ppm levels of O3, it feels like a possibility.

Response: We agree that the flow tube experimental conditions could be different compared to the actual ambient conditions in the oil sands region. Nevertheless, the major reason of reporting the results of our flow tube experiments is to illustrate the large potential of generating pON from the photo-oxidation of bitumen hydrocarbon vapor in the presence of $NO_x$ rather than to conduct a comprehensive mechanistic study of SOA or pON production. Therefore, our flow tube experiment is not designed to fully investigate the detail of individual reaction mechanism. In this work, we report that the pON-to-SOA ratios observed in the flow tube experiments are comparable to those observed from our ambient data, which can be reaction mechanism dependent. To highlight the uncertainties due to the flow tube experimental conditions, the following sentences have been added to the revised manuscript.

Page 13, lines 15-18 "Although our results demonstrate that bitumen hydrocarbon vapor has a large potential to produce pON, it is important to note that our flow tube observations might not fully represent our ambient observations. More research is required to investigate the relative importance of different reaction pathways under various atmospheric conditions that are relevant to the oil sands region."

3. In section 3.3, the loss of pON is suggested several times as a possible or likely mechanism to explain the results of Fig. 3. Based on the combined measurements from the ground, the aircraft, and the laboratory, I am not convinced that there is evidence for pON loss instead of limited pON production at longer timescales. The aircraft and ground data presented in Fig. 3b show an increase in pON concentration with increasing PCA. Furthermore, Lee et al., 2015b found that pON concentrations at another oil and gas producing region could be well described with no additional loss of pON besides turbulent mixing.

Response: In fact, we don't have evidence to rule out the possibilities of either pON loss and limited pON production at longer timescales. Therefore, we would like to list the two possibilities, which are important for future studies on this subject. Nevertheless, we agree that the observations reported by Lee et al., 2015b are important and should be included in this part of discussion as shown below.

Page 14, lines 14-15: "Note that pON concentrations at the oil and gas producing region located at Uintah Basin, Utah, during winter could be well described with no additional loss of pON besides turbulent mixing (Lee et al., 2015)."

To me, this suggests that the apparent decrease in pON with time is due entirely to the normalization to total SOA concentration. More broadly, I would appreciate further discussion about what is gained by framing the results in terms of the fraction of pON or LO-OOA to total SOA, rather than examining the concentration of pON or LO-OOA directly. Put differently, does examining the decreasing trend in pON-to-SOA tell us anything about the production or loss of pON, or does it tell us something about the production of SOA as a whole?

Response: Most of our discussion (for Figures 1 and 2) focuses on the absolute pON mass concentrations observed in the field and aircraft measurements. Figure 3b shows the absolute pON concentrations normalized by BC for illustrating the secondary nature of pON and net pON production as a function of PCA observed in this work. We only use pON-to-LO-OOA and pON-to-SOA ratios in Figure 3c to compare the field and flow tube data. The flow tube reactor is not an ideal approach to determine the pON or SOA formation yield due to the large surface to volume ratio of the reactor and high oxidant exposure within a short residence time compared to conventional smog chamber experiments. Therefore, Figure 3c is primarily used to illustrate the large relative contribution of pON to the SOA mass produced by the oxidation of bitumen vapor, which is comparable to the field observations, rather than to provide yield data and the detail of reaction mechanism of pON production and loss (see also general comment #2). The decreasing trends of pON-to-SOA (or LO-OOA) ratios suggest the decreasing contribution of pON to fresh SOA as a function of PCA, which can be due to a few possibilities as discussed in the original manuscript.

Specific Comments:
Page 3, Line 16-21: I think the results of Lee et al., 2015b should be given more consideration, as that paper found that daytime oxidation of anthropogenic long-chain hydrocarbons was a major source of pON in an oil and gas producing region.

Response: We agree that Lee et al., 2015b is an important literature to highlight the significance of daytime oxidation of anthropogenic long-chain hydrocarbons as a major source of pON in oil and gas producing region. In the original version of this manuscript, this paper has been cited three times in the introduction to highlight the important contribution of this literature to this subject. Furthermore, additional information has been added in the discussion based on the findings of Lee et al. (2015b) (see general comment #3).

Page 5, Line 3-4: See general comment 1 above. I don't feel the connection to urban regions has been adequately made in this manuscript to make this claim.

Response: See the above response to the general comments #1.

Page 5, Line 11: For those unfamiliar with this region, could you clarify the relation between the Alberta oil sands and the Athabasca oil sands?

Response: Athabasca oil sands is the major oil sands region in Alberta. To avoid the confusion, Alberta oil sands region is used throughout the revised version.

Page 9, Line 13-18: The association of different PMF factors with different sources plays a large enough role in this analysis that I think a greater discussion of the different PMF factors and their likely sources would be appropriate to include in the main text.

Response: The focus of this paper is not the source apportionment of organic aerosols in the Alberta oil sands region. A comprehensive investigation of air pollutant sources in Alberta oil sands has been reported in our previous studies with our SP-AMS data being integrated (Tokarek et al., 2018). In order to provide more complete description of other organic aerosol sources, the brief description of PMF factors have been moved to the main text of the revised version (Section 3).

Page 11, Line 6: The description here describes pON mass being well correlated with freshly formed anthropogenic SOA, while Fig. 2a shows total organic aerosol - is the assumption that all the SOA measured is freshly-formed and anthropogenic, or was only a one SOA factor used in the correlation?

Response: Liggio et al. (2016) reported that the oil sands plume was mainly fresh SOA mixed with the background SOA being observed outside the plume. The correlation is determined using the fresh SOA factor (i.e. total OA subtracted by the background SOA factor reported by Liggio et al. (2016)). This information has been clearly stated in the revised version to avoid confusion.

Page 11, lines 18-21: "The pON mass derived from these aircraft measurements correlated well with anthropogenic SOA that was freshly formed during plume dispersion (i.e., both pON and fresh SOA are background subtracted as reported in Liggio et al. (2016)) ($r^2 = 0.71$, Figures 2a, 2b and S13), and accounted for up to ~34 % (±18%) of such fresh SOA by mass."

Page 11, Line 23: I parse this sentence as saying that because LO-OOA was strongly associated with unresolved IVOCs, the IVOCs must be produced by oil and gas extraction activities, which does not make sense to me. This reasoning should be clarified or an alternate justification for the likely sources of IVOCs should be provided.

Response: We agree that the original interpretation is not clear. Source identification of IVOCs for this field campaign has been discussed in detail in Tokarek et al. (2018). The potential sources of IVOCs listed in this manuscript are referred to the results from Tokarek et al. (2018). The paragraph has been revised as following:

Page 12, lines 11-16: "Tokarek et al. (2018) performed principle component analysis of 28 co-located measurements at the ground site and reported that LO-OOA was strongly associated with unresolved IVOCs measured by gas chromatograph-ion trap mass spectrometry (GC-ITMS), while biogenic VOCs including α-pinene, β-pinene and limonene were not associated with LO-OOA. Their results of principle component analysis further illustrated that tailings ponds, mine fleet and vehicle emissions, and mining and processing of raw bitumen are the major sources of IVOCs in the oil sands region."

Page 12 Line 21 - Page 13, Line 7: I am having trouble understanding how this para-graph fits into the chain of reasoning in section 3.2. The previous paragraphs form a seem to follow a logical progression, setting up oil-sands IVOCs as the likely precursor for pON and confirming with flow tube experiments. I would suggest clarifying the goal of this paragraph, or moving it to a different section in the text.

Response: The primary goal of this paragraph is to highlight the fact that hydrocarbons are the major SOA precursors emitted from oil sands operations (Li et al., 2017). Based on previous laboratory investigations, the pON formation potential of hydrocarbons depends on their molecular structures and reaches a plateau for SOA precursors in the IVOCs range. We have shortened this paragraph and combined it with the former paragraph to make the information more connected in this section.

Page 13, lines 10-14: "Note that the pON formation potential of hydrocarbons depend on their molecular structures. Matsunaga and Ziemann (2010) demonstrated that yields of pON due to OH radical oxidation in the presence of $NO_x$ increased with the carbon number of 2-methyl-1-alkenes (from $C_9$ to $C_{15}$), primarily due to enhanced gas-to-particle partitioning, and reached a plateau for $C_{14}$-$C_{15}$ precursors that fall into the IVOCs range of hydrocarbons observed in the bitumen vapour as discussed above."

Page 13, Line 10-11: How does the analysis of photochemical age for the ground measurements distinguish between aging of an isolated air mass and mixing of air masses of different ages from different sources? Is this what the normalization to black carbon was used for? If so, this should be stated explicitly in the text.

Response: The analysis of photochemical age cannot distinguish between the two types of air masses mentioned by the reviewer. In particular, the relatively low pON-to-rBC ratio when the PCA was high can be due to the combined effects of atmospheric aging and mixing of air masses. This information has been added to the revised manuscript as shown below.

Page 14, lines 6-8: "Relatively low $NO_x$ levels might limit pON production when the estimated PCA > 1 h (Figure 3d, i.e., at times without strong influence from $NO_x$ emissions) but the effects of atmospheric mixing of aged background air masses cannot be ruled out."

Figure 3: It might be useful to include in this figure or in the supplement how the absolute concentration of pON changed with photochemical age in the flow tube experiment. These results might be able to help clarify whether loss of pON was being observed.

Response: Thanks for the suggestion. Since the flow tube experiment is not designed for determining the SOA and pON yield, the absolute changes in the concentration of pON are not discussed in this manuscript (please see the detail in the response for the general comments #3).

Page 13 Line 21 - Page 14 Line 2: This sentence appears to equate the evaporation or degradation of pON with changes to pON production. This should be reworded to clarify that the production and loss terms of pON can change independently. I am also confused by the reference to Figure 1k, as that figure does not directly reference photochemical age at all.

Response: We agree with the reviewer that the production and loss terms of pON can change independently as a function of PCA. To avoid the potential confusion, the sentence has been revised to clarify that the loss terms limit the net production of pON observed in this work.

Page 14, lines 11-13: "Substantial evaporative loss (i.e., LO-OOA generally represents a more volatile fraction of OOA), hydrolysis and photo-degradation of pON when both temperature and photochemical activity were high might limit the net pON production at longer PCA."

Page 15, Line 11-14: Could this sentence be re-worded? I am not sure that it is grammatically correct and it took me several attempts to understand its meaning, that the production of pON from the Alberta oil sands is significant on regional or continental scales.

Response: This sentence has been revised as shown below:

Page 16, lines 1-4: "Given the recent observation that oil sands operations in Alberta can be one of the largest anthropogenic sources of SOA in North America (45–84 tonnes/day) by comparing to the estimated SOA production rates in different cities (Liggio et al., 2016), the production of pON in the oil sands pollutant plumes can be significant on regional or continental scales."

Technical Corrections:
Page 3, Line 24: The last sentence on this page ('Until recently....') is not grammatical.

Response: This has been revised as "Recently, …"

Page 7, Line 21: The citation 'Liggio et al. (2016)' appears twice.

Response: One of the citations was removed.

Page 14, Line 14-16:  This clause ('which may make the formation of non-pON fresh SOA') is missing an object.

Response: The sentence has been revised as below.

"Such decreasing trend could be due to decreasing mixing ratios of NOx (Figure 3d) in the plume caused by dilution downwind of the emission source, which may make the formations of non-pON fresh SOA and multi-generation products more important in the later stage of oxidative process,…"

**Reviewer 2:**

The manuscript presents ground and flight-based observations of particulate organic nitrates and total organic aerosol mass concentrations in the summer-fall season in/near the Athabasca oil sands region. According to the calculations presented, pON contributes as much 55% of OA mass concentration in freshly emitted air mass, but that contribution decreases with photochemical age. This reviewer has serious concerns regarding the assumptions made in (1) quantifying pON, and (2) attributing the field observed enhancement in OA to oxidation of bitumen vapors. Major revisions/clarifications are needed prior to publication.

(1) A clearer accounting of pON mass contribution is needed. As the authors note, figure 3 shows the upper limit contribution of pON by assuming a constant molecular weight of ON of 300 g/mol. How representative is this value to ON derived from bitumen vapor oxidation? The range of typical ON molecular weights reported by Farmer et al. & Xu et al. were focused on different parent VOCs. Moreover, a wide range in R values was observed during the campaign, as seen in figure S8. Doesn't this reflect that numerous parent compounds contributed to pON growth, hence, of varying molecular weight of pON? Is this variability in R accounted for in the quantification of pON? If not, show how big an impact the variability in R could have on pON quantification. Is there a trend in the R value with photochemical age? Could the changing R with PCA be responsible for the apparent decreasing pON/OA ratio? Revise figure 3 and numbers reported in the manuscript to clearly show a reasonable range of pON contribution to OA by acknowledging the assumptions made here.

Response: As discussed in Section 3.1 (Table S3) and the experimental section, the pON mass concentrations are calculated based on different combinations of $NO^+/NO_2^+$ ratios of ON (i.e. $R_{ON}$ = 5 and 10) and molecular weight of pON (i.e., $MW_{ON}$ = 200-300 g/mol) due to the fact that $R_{ON}$ and $MW_{ON}$ were unlikely constant in the field study as pointed out by the reviewer. Therefore, the lower and upper limits of the calculated pON mass concentrations are reported. To make it clear in the revised manuscript, a sentence has been revised in Section 3.1 as shown below:

Page 11, lines 9-12: "A strong correlation between the mass loadings of pON and LO-OOA ($r^2$ = 0.77) suggests that ON was a component of LO-OOA, contributing approximately 24-53% of LO-OOA mass (i.e., represents the lower and upper limits based on different combination of $NO^+/NO_2^+$ ratios (i.e., 5 and 10) and molecular weight of pON (i.e., 200-300 g/mol) in the calculation) (Figure S9 and Table S3)."

The temporal variations of pON determined from different scenarios are very similar to each other. Therefore, only the case of $R_{ON}$ = 5 and $MW_{ON}$ = 300 g/mol (i.e., represents the upper limit of pON mass) is selected to illustrate the changes in pON/rBC and pON/OOA (panel b and c in Figure 3, respectively) ratios as a function of photochemical age (PCA). The major conclusions from these two panel would not be affected. Table S3 shows the pON mass concentrations based on different $R_{ON}$ and $MW_{ON}$.

Also, it is noted on line 7-9 page 7 that R = 3.5+/-1.5 was determined by calibrations, but no details of these calibrations are provided, and an R value of 5 is used in the figures. How many different

VOCs (aromatics, alkanes, alkenes, etc.) were characterized? What governs the variability in the R values?

Responses: Ammonium nitrate (NH4NO3) particles generated by a constant output atomizer (TSI Inc., Model 3076) were dried using a diffusion dryer, and were subsequently size selected at 300 nm using a differential mobility analyzer (DMA, TSI Inc., Model 3081) for determining the mass-based ionization efficiency of nitrate ($mIE_{NO3}$) and the ionization efficiency of ammonium relative to nitrate ($RIE_{NH4} = mIE_{NH4}/mIE_{NO3}$) when the instrument was operated in the laser-off mode. This information can be found in the supplement information.

The $R_{ON}$ values can be pON species dependent so that a range of $R_{ON}$ values were used in the estimation. The assumption of $R_{ON}$ values and molecular weights of pON are based on Farmer et al. (2010). So far, there is no literature available for pON generated by IVOCs.

(2) The authors use the fact that similar fractional contributions of pON to SOA – that is, between 30 and 55% - were observed in the ambient fresh plumes as in the flow tube experiments as evidence that bitumen vapors were the source of the pON (and SOA). Using such a metric - particularly one with a sizeable range, in a flow tube with unrealistic chemical conditions, where pON contribution to OA appears strongly dependent on photochemical lifetime - as an identifying marker seems highly questionable as it most certainly will not be specific to bitumen vapor oxidation. Have the authors attempted any other VOCs - say isoprene, monoterpenes, or any of the possible emission sources listed at the end of page 11 and start of page 12 - in the flow tube experiments to rule out other VOC sources?

Response: Thanks for the comments. We are not using the pON-to-SOA ratio from the flow tube experiments to confirm IVOCs are the major precursors of pON. Instead, Liggio et al. (2016) showed that IVOCs as the major sources of the fresh SOA in the oil sands emission plumes, and in this work we find that pON is a large contributor to such fresh SOA in the oil sands region based on both ground and aircraft measurements. We agree that the flow tube experimental conditions may not be atmospherically relevant but the results can be used to demonstrate the pON formation potential from the bitumen vapor oxidation. We believe our finding still can provide important insight into pON formation from IVOCs originally from anthropogenic emissions.

There is an odd sentence in the conclusion, that "pON accounted for 21% of total OA mass, which is comparable to other locations," studies by Kiendler-Scharr et al. and Ng et al. Where did this 21% come from? Also, both of those studies focused on residential and urban areas. But if pON/OA is comparable regardless of region of study, why should we care about pON from oil sands? Isn't it possible the plumes intercepted by the aircraft had elevated HOx (due to elevated NOx) that rapidly oxidized biogenic VOC entrained into the plume?

Response: This is the campaign averaged mass contribution of pON to total OA (i.e. = HOA + LO-OOA and MO-OOA in this field study) for comparing results from other locations. Even though the overall mass fraction contribution of pON to OA from oil sands are comparable to other locations, the high pON-to-LO-OOA ratio (30-55%) highlights the significance of pON in the

SOA production within in the oil sands emissions. Furthermore, the large production of anthropogenic SOA from within the oil sands pollution plumes make it becomes an important source of pON in regional scale (Liggio et al., 2016). Liggio et al. (2016) have also shown that biogenic VOC were not the major precursors to produce SOA within the plumes.

The authors conclude the pON is formed largely by daytime chemistry. I would like to see included in figure 1 a diel plot of the fraction of pON to OA.

Response: Figure 1k clearly shows that the absolute pON and LO-OOA mass concentrations peaked during the daytime. In addition, the aircraft measurement was conducted during daytime to measure the OA composition with the pollution plumes. Instead of adding extra panel in Figure 1, the diurnal plot of pON-to-OA ratio has been added to the SI (Figure S14). Note that the ratios are not only affected by the pON formation chemistry but also OA components from other sources.

[Figure]

**Figure S14**: Diurnal plot of pON/total OA ratio. The upper and lower values of shaded regions represented 25 and 75 percentiles of diurnal variations, respectively.

Minor comments:
Page 10 line 20-22. the 24-53% range, is that accounting for variability through campaign assuming a constant molecular weight, or range due to assuming 200-300g/mol molecular weight?

Response: As reported in Table S3, the reported range (24-53%) represents the lower and upper limits, which is determined based on different combinations of $NO^+/NO_2^+$ ratios (i.e. 5 and 10) and molecular weight (i.e., 200-300 g/mol) in our calculation. Therefore, the lower and upper limits of the calculated pON mass concentrations are reported. The sentence has been revised as following to avoid the confusion.

Page 11, lines 9-12: "A strong correlation between the mass loadings of pON and LO-OOA ($r^2 =$ 0.77) suggests that ON was a component of LO-OOA, contributing approximately 24-53% of LO-OOA mass (i.e., represents the lower and upper limits based on different combination of $NO^+/NO_2^+$ ratios (i.e., 5 and 10) and molecular weight of pON (i.e., 200-300 g/mol) in the calculation) (Figure S9 and Table S3)."

Page 11 line 5-7; this sentence is not supported by the preceding sentence. Are you including the mass of the nitrate functional group when reporting mass of pON or just the organic portion?

Response: The mass of nitrate functional group is included in the total mass of pON based on our calculation approach. The two sentences have been revised as following:

Page11, lines 16-21: "A higher average value of $NO^+/NO_2^+$ ratio of SOA-rich plumes compared to that of pure ammonium nitrate was also observed in our aircraft measurements, further demonstrating the presence of pON in SOA. The pON mass derived from these aircraft measurements correlated well with anthropogenic SOA that was freshly formed during plume dispersion (i.e., both pON and fresh SOA are background subtracted as reported in Liggio et al. (2016)) ($r^2 = 0.71$, Figures 2a, 2b and S13), and accounted for up to ~34 % ($\pm$18%) of such fresh SOA by mass."

Page 3 lines 12-14, is that true, that the composition of pON can affect SOA growth and npf? In any case, these are probably not the correct citation.

Response: The sentence has been revised as shown below.

Page 3, lines 12-15: "Furthermore, pON can be highly functionalized, which has been observed in the events of new particle formation and secondary organic aerosol (SOA) growth (Ehn et al., 2014;Lee et al., 2016), with strong impacts on air quality and climate (Hallquist et al., 2009;Kanakidou et al., 2005)."

Page 4 lines 19 - 21, need citation

Response: This sentence highlights the key finding of this study rather than the literature information. This sentence has been revised as following to avoid confusion.

Page 4, lines 19-21: "Our ambient and laboratory measurements illustrate that the observed pON production and the relative importance of pON to the freshly formed SOA depend upon the degree of photochemical aging in the polluted atmosphere."

**Interactive comments from Atkinson**

Very nice work! In pondering the apparent decrease in the pON content (and relative fraction of SOA) with PCA that you report, I was reminded of another recent paper that is still in the open discussion (I think). The paper by Schwantes, et al. https://doi.org/10.5194/acp-2018-1358 reports the formation of organic dinitrates (among other products) during the high-NOx photooxidation of isoprene and suggests (section 5.3) a negative measurement bias in the AMS determination of these com-pounds (and perhaps all pON). It isn't clear to me that the methods used to determine pON are the same in both papers, but it begs the question of whether some sort of interference could be affecting the measurements of the highly oxidized oil sands SOA, perhaps leading to a lower effective CE for the more oxidized dinitrates. Perhaps this could account for the apparent decrease in the absolute and relative contribution of the nitrated products

Response: Thanks for pointing out this issue. Schwantes et al. (2019) suggests that the collection efficiency (CE) and/or the ionization efficiency of the AMS is possibly lower for isoprene-SOA that are dominated by low-volatility nitrates and dinitrates reaction pathways. Based on their laboratory observations and previous field observations, they suggested that further AMS calibration of organic nitrates is necessary. Although their observations may not be able to generalize for pON generated by other types of precursors, we agree that it is worth mentioning this recent finding in the revised version.

Page 16, lines 4-8: "Schwantes et al. (2019) has reported that collection efficiency and/or relative ionization efficiency of pON produced via photo-oxidation of isoprene was lower than those of other isoprene-SOA products in AMS measurements. Although such measurement uncertainty has not been generalized for pON generated from other SOA precursors, underestimation of pON mass contribution to total SOA is possible in this work."

**References:**

Eluri, S., Cappa, C. D., Friedman, B., Farmer, D. K., and Jathar, S. H.: Modeling the Formation and Composition of Secondary Organic Aerosol from Diesel Exhaust Using Parameterized and Semi-Explicit Chemistry and Thermodynamic Models, Atmos. Chem. Phys. Discuss., 2017, 1-30, 10.5194/acp-2017-1060, 2017.

Farmer, D. K., Matsunaga, A., Docherty, K. S., Surratt, J. D., Seinfeld, J. H., Ziemann, P. J., and Jimenez, J. L.: Response of an aerosol mass spectrometer to organonitrates and organosulfates and implications for atmospheric chemistry, Proceedings of the National Academy of Sciences of the United States of America, 107, 6670-6675, 10.1073/pnas.0912340107, 2010.

Hodzic, A., Kasibhatla, P. S., Jo, D. S., Cappa, C. D., Jimenez, J. L., Madronich, S., and Park, R. J.: Rethinking the global secondary organic aerosol (SOA) budget: stronger production, faster removal, shorter lifetime, Atmospheric Chemistry and Physics, 16, 7917-7941, 10.5194/acp-16-7917-2016, 2016.

Lee, L., Wooldridge, P. J., deGouw, J., Brown, S. S., Bates, T. S., Quinn, P. K., and Cohen, R. C.: Particulate organic nitrates observed in an oil and natural gas production region during wintertime, Atmospheric Chemistry and Physics, 15, 9313-9325, 10.5194/acp-15-9313-2015, 2015.

Li, S. M., Leithead, A., Moussa, S. G., Liggio, J., Moran, M. D., Wang, D., Hayden, K., Darlington, A., Gordon, M., Staebler, R., Makar, P. A., Stroud, C. A., McLaren, R., Liu, P. S. K., O'Brien, J., Mittermeier, R. L., Zhang, J. H., Marson, G., Cober, S. G., Wolde, M., and Wentzell, J. J. B.: Differences between measured and reported volatile organic compound emissions from oil sands facilities in Alberta, Canada, Proceedings of the National Academy of Sciences of the United States of America, 114, E3756-E3765, 10.1073/pnas.1617862114, 2017.

Liggio, J., Li, S. M., Hayden, K., Taha, Y. M., Stroud, C., Darlington, A., Drollette, B. D., Gordon, M., Lee, P., Liu, P., Leithead, A., Moussa, S. G., Wang, D., O'Brien, J., Mittermeier, R. L., Brook, J. R., Lu, G., Staebler, R. M., Han, Y. M., Tokarek, T. W., Osthoff, H. D., Makar, P. A., Zhang, J. H., Plata, D. L., and Gentner, D. R.: Oil sands operations as a large source of secondary organic aerosols, Nature, 534, 91-94, 10.1038/nature17646, 2016.

Liu, H., Man, H., Cui, H., Wang, Y., Deng, F., Wang, Y., Yang, X., Xiao, Q., Zhang, Q., Ding, Y., and He, K.: An updated emission inventory of vehicular VOCs and IVOCs in China, Atmos. Chem. Phys., 17, 12709-12724, 10.5194/acp-17-12709-2017, 2017.

Schwantes, R. H., Charan, S. M., Bates, K. H., Huang, Y., Nguyen, T. B., Mai, H., Kong, W., Flagan, R. C., and Seinfeld, J. H.: Low-volatility compounds contribute significantly to isoprene secondary organic aerosol (SOA) under high-NOx conditions, Atmos. Chem. Phys., 19, 7255-7278, 10.5194/acp-19-7255-2019, 2019.

Tokarek, T. W., Odame-Ankrah, C. A., Huo, J. A., McLaren, R., Lee, A. K. Y., Adam, M. G., Willis, M. D., Abbatt, J. P. D., Mihele, C., Darlington, A., Mittermeier, R. L., Strawbridge, K., Hayden, K. L., Olfert, J. S., Schnitzler, E. G., Brownsey, D. K., Assad, F. V., Wentworth, G. R., Tevlin, A. G., Worthy, D. E. J., Li, S. M., Liggio, J., Brook, J. R., and Osthoff, H. D.: Principal component analysis of summertime ground site measurements in the Athabasca oil sands with a

focus on analytically unresolved intermediate-volatility organic compounds, Atmos Chem Phys, 18, 17819-17841, 10.5194/acp-18-17819-2018, 2018.

Zhao, Y. L., Hennigan, C. J., May, A. A., Tkacik, D. S., de Gouw, J. A., Gilman, J. B., Kuster, W. C., Borbon, A., and Robinson, A. L.: Intermediate-Volatility Organic Compounds: A Large Source of Secondary Organic Aerosol, Environmental Science & Technology, 48, 13743-13750, 10.1021/es5035188, 2014.

Zhao, Y. L., Nguyen, N. T., Presto, A. A., Hennigan, C. J., May, A. A., and Robinson, A. L.: Intermediate Volatility Organic Compound Emissions from On-Road Diesel Vehicles: Chemical Composition, Emission Factors, and Estimated Secondary Organic Aerosol Production, Environmental Science & Technology, 49, 11516-11526, 10.1021/acs.est.3b02841, 2015.